# Single-Cell Analysis of Host Responses in Bovine Milk Somatic Cells (bMSCs) Following HPAIV Bovine H5N1 Influenza Exposure

**DOI:** 10.3390/v17060811

**Published:** 2025-06-03

**Authors:** Gagandeep Singh, Sujan Kafle, Patricia Assato, Mankanwal Goraya, Igor Morozov, Juergen A. Richt

**Affiliations:** 1Department of Diagnostic Medicine/Pathobiology (DMP), College of Veterinary Medicine, Kansas State University, Manhattan, KS 66506, USA; gdsdhol@vet.k-state.edu (G.S.); skafle@vet.k-state.edu (S.K.);; 2Department of Plant Pathology and Environmental Microbiology, The Pennsylvania State University, University Park, PA 16802, USA; mfg5907@psu.edu

**Keywords:** bovine H5N1, single-cell RNA-seq, bovine milk somatic cells

## Abstract

The 2024 outbreak of highly pathogenic avian influenza virus (HPAIV) H5N1 in U.S. dairy cattle presented an unprecedented scenario where the virus infected bovine mammary glands and was detected in milk, raising serious concerns for public health and the dairy industry. Unlike previously described subclinical influenza A virus (IAV) infections in cattle, H5N1 infection induced severe clinical symptoms, including respiratory distress, mastitis, and abnormal milk production. To understand the host immune responses and changes, particularly in the mammary gland, we performed single-cell RNA sequencing analysis on bovine milk somatic cells (bMSCs) in vitro exposed to an H5N1 isolate from an infected dairy farm. We identified ten distinct cell clusters and observed a shift toward type-2 immune responses, characterized by T cells expressing *IL13* and *GATA3*, and three different subtypes of epithelial cells based on the expression of genes associated with milk production. Our study revealed temporal dynamics in cytokine expression, with a rapid decline in luminal epithelial cells and an increase in macrophages and dendritic cells, suggesting a role in increased antigen presentation. While viral RNA was detected in bulk-exposed bMSC samples via qRT-PCR, no viral reads were observed in the scRNA-seq data, indicating that the immune responses captured may be due to exposure to viral components rather than productive infection. This research fills a critical gap in understanding the immune responses of bovine mammary glands to H5N1 exposure and highlights the need for further investigation into therapeutic strategies for managing such outbreaks.

## 1. Introduction

Highly pathogenic avian influenza viruses (HPAIVs) have predominantly affected avian species, but their capacity to spill over into mammalian hosts poses a significant public health and agricultural concern [1,2,3]. The 2024 outbreak of HPAIV H5N1 in United States dairy cattle represents a novel and alarming example where the virus has infected cattle, specifically targeting cow mammary glands [1,2,3]. Unlike previous influenza A virus (IAV) infections in cattle, which are generally subclinical, the H5N1 outbreak was associated with severe clinical symptoms, including respiratory distress, reduced feed intake, abnormal milk production, and mastitis with elevated somatic cell counts [1,2,3]. More concerning is the detection of high loads of infectious viral particles in cow milk, highlighting potential risks for viral transmission to humans through dairy products [1,2,3].

Cattle with bovine H5N1 infection display neutrophilic and lymphoplasmacytic mastitis, which mirrors the immune response seen in bacterial infections of the udder [1,2,3,4,5]. This results in severe inflammation and structural damage to the mammary glands. While bovine mastitis has been extensively studied in the context of bacterial infections, there is limited understanding of how viral infections, particularly by IAV, modulate the mammary gland’s immune responses. To address this gap, we employed single-cell RNA sequencing (scRNA-seq) to profile milk somatic cells exposed to bovine H5N1 to better understand the bovine mammary gland state during bovine H5N1 exposure. We observed significant changes in luminal epithelial cells, with distinct cytokine expression patterns reflecting the temporal dynamics of the immune response.

## 2. Materials and Methods

### 2.1. Cells and Viruses

Madin–Darby Canine Kidney (MDCK, ATCC, cat# CCL-34) cells were cultured as routine in Dulbecco’s Modified Eagle Medium (DMEM) at 37 °C, supplemented with 10% fetal bovine serum (FBS) and a 1× antibiotic–antimycotic solution (ThermoFisher Scientific, Waltham, MA, USA).

The bovine H5N1 HPAIV (A/dairy cattle/Kansas/5/2024) was isolated from a HPAIV-infected dairy farm in Kansas and propagated in 9-day-old SPF chicken eggs and plaque titered using MDCK cells, as described previously [6].

### 2.2. Raw Milk Processing and Collection of Bovine Milk Somatic Cells (bMSCs)

Fresh raw milk (50 mL from each quarter) from three multiparous Holstein Friesian cows was collected via hand milking at the Kansas State University Dairy Teaching and Research Center (Manhattan, KS, USA). Milk was collected directly into the collection vial during forestripping, and milk was collected from each quarter of the udder and composited. The collected raw milk was centrifugated at 400× *g* for 10 min at 4 °C, the supernatant and fat were removed, and the pellet was washed twice with chilled (4 °C) RPMI growth media (1× RPMI 1640+ 2% FBS + 1× antibiotic–antimycotic; Thermos Scientific, Waltham, MA, USA). The viable bMSCs were enriched using a Dead Cell Removal Kit (Miltenyi Biotec, Bergisch Gladbach, Germany) and bMSCs were resuspended in warm (37 °C) RPMI growth media and pooled together for virus infection.

### 2.3. bMSCs’ Exposure to HPAIV Bovine H5N1

Pooled bMSCs plated at a density of 1 × 10^6^ cells/well in a Ultra-Low attachment 6-well plate (Corning) were exposed to 0.01 multiplicity of infection (MOI) with either bovine H5N1 or infection media. The cells and viruses were then incubated together for 24 h at 37 °C supplemented with 5% CO_2_. After 24 h incubation, cells were centrifuged at 400× *g* for 5 min at 4 °C to remove debris and ambient host and viral DNA/RNA. The cells were again subjected to the Dead Cell Removal Kit (Miltenyi Biotec) for the removal of dead cells and debris and to reach the viable cell target of more than 70% live cells for scRNA-seq experiments.

### 2.4. Single-Cell Sequencing Using 10X Genomics Platform

Single-cell suspensions from both control and exposed samples processed as above were subjected to single-cell RNA-seq (scRNA-seq) using Chromium Next GEM Single Cell 5’ Reagent Kits v2 (10× genomics) following the manufacturer’s instructions. Briefly, single-cells suspensions were loaded onto a Chromium Single Cell Chip (10× Genomics) for co-encapsulation with barcoded Gel Beads at a target capture rate of ~10,000 individual cells per sample using the Chromium Controller (10× Genomics). The co-encapsulation cells with barcoded Gel Beads were subjected to RT-PCR cycles for barcoding the capture cell’s mRNA and cDNA synthesis. Synthesized cDNA was brought out of BSL-3 biocontainment using approved protocols. The cDNA was further indexed and processed for Illumina sequencing as per the manufacturer’s instructions. The indexed samples were pooled together with equal concentrations of DNA molecules and sequenced on an Illumina NextSeq 550 as per the 10× Genomics instructions.

### 2.5. Bioinformatics Analysis

The sequencing process yielded raw fastq files which were used for input for the cellranger package (10× Genomics) for scRNA-seq analysis [7]. To quantify and align reads to a host and virus genome, the bovine reference and annotation (ARS-UCD1.2/bosTau9) was combined with A/Dairy Cattle/Kansas/5/(NCBI GenBank ID PP732373-80) genome and annotation files. The resulting aligned and analyzed files were visualized and further analyzed using Loupe Brower (10× Genomics) as follows: The processed data were cleaned up by removing cells with fewer than 5000 reads and a unique molecular identifier (UMI) barcode, followed by re-clustering using 50 principal components in PCA and 50 neighbors for a t-distributed stochastic neighbor embedding (t-SNE) plot [8]. (Figure 1). Identification of cells in the cultures was performed by a literature search and the reporting of gene markers for single-cell types [9,10,11,12]. Each cluster was then re-named based on the cell type.

Differential gene expression between the cluster as well as samples was performed using Loupe Brower (10× Genomics) [7]. The resulting gene matrixes were visualized using pheatmap [13]. The top 50 upregulated genes after infection were further analyzed for gene function enrichment using the Metascape website [14]. To quantify the cell number for each cluster in each sample, individual clusters were selected and then split into sample IDs. The resulting count matrix for cells was then visualized using Prism v10.2.2 (GraphPad).

### 2.6. RNA Extraction and qPCR for Viral RNA Detection

The RNA for qRT-PCR for the detection of viral RNA was extracted from infected/exposed cells using automated bead extraction as previously described [6]. RNA extracted from bovine H5N1-inoculated bMSCs and MDCK cells were subjected to a one-step RT-qPCR targeting the Influenza A Matrix gene [6].

## 3. Results

### 3.1. Bovine Milk Somatic Cells (bMSCs) Consist of Diverse Luminal and Immune Cells

Typically, milk from healthy cattle consists of approximately 100,000 cells/mL with 30–40% viability, which is low for scRNA-seq experiments [15,16]. To overcome low viability, we enriched our collected cells three times, two times before exposure and one time after exposure, by removing dead cells and pooling the cells from three animals. A total of 10,000 pooled cells/treatment were analyzed using the 10× Genomics controller and cellranger pipeline, from which ~6900 cells/treatment were recovered with ~16,500 mean reads/cell [7].

The control and bovine H5N1-exposed cells were initially analyzed individually using the 10× Genomics cellranger pipeline [7]. All data were merged together, unsupervised clustering performed using 50 principal components for Principal Components Analysis (PCA), and cells projected on a two-dimensional t-distributed stochastic neighbor embedding (t-SNE) plot [8] (Figure 1). We identified 10 cell clusters and utilized the 28 selected gene markers curated from various published databases to identify the cell type of the individual cluster [9,10,11,12] (Figure 1A,B). Seven clusters, constituting ~70% of cells, were identified as immune cells with 48% as monocytes, 10% as T cells, 7% as macrophages, and 3% as dendritic cells (DCs) (Appendix A). The other three clusters, constituting ~30% cells, were characterized as luminal epithelial cells with 13% as subset 1 expressing low casein genes, 11% as subset 2 expressing medium casein genes, and 7% as subset 3 with high casein gene expression (Figure 1B, Appendix A).

Differential gene expression from the scRNA-seq data further resolved the cell subsets and their functional states (Figure 1, Appendix A). Monocytes subset 1 was comprised of cells expressing genes *PKD2L1*, *CACNA2D3*, and *ENO4* (corresponding to classical monocytes); monocytes subset 2 consisted of cells expressing *CCDC42* and *ZNF713* (corresponding to non-classical monocytes); and monocytes subset 3 was comprised of cells expressing genes *LAMA3, RNASE12,* and *RTKN2* (corresponding to intermediate monocytes) [9,10,11,12]. The macrophage cluster consisted of cells expressing genes *CLDN1, CYP1B1, MMP1, MMP3, RBP4, CSF3, MMP12,* and *HMGA2* which were upregulated with inflammatory stimuli [17,18,19,20,21,22,23]. The DC cluster was comprised of cells expressing genes *MGP, GARNL3,* and *RARRES2* (corresponding to plasmacytoid DCs) [24]. T-cell subset 1 contained cells expressing *CCL1, IL13, LTA, TNFRSF4, KLRB1,* and *TP73* genes (corresponding to activated Th_2_ T cells) [25,26,27,28,29,30,31,32], whereas T-cell subset 2 expressed genes *EGFL8, FBXO44,* and *IL-15L* corresponding to inhibited Th_2_ T cells [33,34,35,36]. Luminal cell subset 1 was comprised of cells expressing genes *ISM1, CCL19* (leukocyte chemoattractant), *CLEC1A*, and Ubiquitin-like protein 5 (corresponding to stress response apoptosis-inducing protein) [37,38,39,40,41,42]. Luminal cells subset 2 consisted of cells expressing genes *ENTPD3, SRRM3,* and *GPM6A* (corresponding to breast cancer-related genes) [43,44,45]. Luminal cells subset 3 was comprised of cells expressing *DNAH9, PTPRH,* and *DYDC1* genes (corresponding to ciliated luminal epithelial cells) [46,47,48].

### 3.2. Bovine H5N1 Exposure Alters the bMSCs’ Cellular Diversity and Their Functional State

Bovine H5N1 has a distinct tropism for epithelial cells lining the alveoli of the udder of cows [1,2]. To assess how bovine H5N1 exposure affects bMSCs, which consist of exfoliated epithelial cells from udder alveoli, we compared the scRNA-seq data from bMSCs, either control or bovine H5N1 exposed, after 24 h post-inoculation [49,50] (Figure 2). The 24 h time point was selected based on viral kinetics observed in MDCK cells, where viral replication peaks without extensive cell death, allowing for optimal viral detection and maintaining cell viability. We combined the reference genomes and annotations for both the bovine genome and the H5N1 virus genome to identify viral reads within the cells; however, no viral RNA reads were observed in any of the cells analyzed. To validate this, we performed M gene-based qRT-PCR on bovine H5N1-inoculated bMSCs and MDCK cells (as a positive control) and successfully detected viral RNA in both cell types (Figure 3). Therefore, the detection of viral RNA by qRT-PCR suggests that some bMSCs may have internalized H5N1 viral material; however, in the absence of viral RNA reads in the scRNA-seq data, we cannot confirm whether any cells were truly infected or supported productive viral replication. We then quantified the number of cells in each cluster and compared them with or without bovine H5N1 exposure. The dynamics of the cell population changed with the virus exposure with increased immune cells and decreased luminal cells in bovine H5N1-exposed cells. Only subset 1 of monocytes and luminal cells remained the same in the control and the bovine H5N1-exposed cell populations. We assessed the change in functional state of cells by differential gene expression followed by Gene Ontology (GO) enrichment of the top 50 upregulated genes from each cell cluster after bovine H5N1 exposure. The monocytes subset 1, which corresponds to classical monocytes, showed upregulation of inflammatory responses with negative regulation of the protein phosphorylation process after infection. The luminal cells subset 1, which has high expression of leukocyte chemoattractant, showed upregulation of the nucleoside triphosphate biosynthetic process and protein polyubiquitination. The GO analysis on subsets 2 and 3 of monocytes revealed upregulation of the pathways related to the cell cycle, macroautophagy, and regulation of the viral process. In macrophages, upregulation of proteolysis involved in the protein catabolic process after infection was observed. The GO analysis also indicated that bovine H5N1 exposure activates the dendritic cell and T-cell subsets by upregulating the cellular responses to cytokines and inflammatory stimulations. Subsets 2 and 3 of luminal epithelial cells, which have higher expression of casein proteins, appeared to be negatively affected by exposure with a decrease in their number and the upregulation of apoptosis pathways (Figure 2, Appendix A).

We further assessed the immunological state of the cells by selecting genes related to Th_1_, Th_2_, Th_9_, and Th_17_ cytokines (Appendix A). Monocytes and dendritic cells showed strong upregulation of pro-inflammatory cytokine genes such as *CSF2, IL1B*, and *TNF*. Subset 3 of luminal cells showed high expression of *IL1B* and *TNF* in contrast to subset 2 of luminal cells, which showed increased expression of the genes *IFNG* and *TGFB*. T cells (subsets 1 and 2) displayed differential expression patterns, with subset 2 showing increased expression of *IFNG, FOXP3*, and *TGFBR1/2* (corresponding to regulatory T cells), while subset 1 showed markers of Th_17_-related responses such as *RORC* and *IL17A*. Macrophages show notable expression of *MMP1*, *MMP3*, and *MMP12*, which are involved in tissue remodeling.

## 4. Discussion

The 2024 outbreak of highly pathogenic avian influenza virus (HPAIV) H5N1 in US dairy cattle is unprecedented due to its unique ability to infect cow mammary glands, with high amounts of infectious viral particles excreted in milk [1,2,3]. This raises concerns for public health and food safety. Unlike previously described subclinical IAV infections in cattle, H5N1 infection leads to severe symptoms, including reduced feed intake, respiratory distress, abnormal milk production, and mastitis with elevated somatic cell counts [1,2,3]. In healthy cattle, somatic cells contain exfoliated udder alveolar epithelial cells and immune cells, whereas in cattle with bacterial mastitis, the number of epithelial cells are diminished and the number of neutrophils significantly increase [4,5,16]. Bovine H5N1 infection in the udder exhibits neutrophilic and lymphoplasmacytic mastitis similar to bacterial infection, associated with severe inflammation and structural damage in the mammary glands [1,2,3]. These findings underscore the need for further studies to understand the role of mammary gland immune responses against viral infections.

Here, we employed scRNA-seq to profile immune cells in the bMSC to explore the state of the udder during bovine H5N1 exposure. One significant challenge in scRNA-seq of MSCs is the low viability of cells, which is further exacerbated in in vitro environments [16]. To overcome this, we removed dead cells at multiple stages of the experiment. Another major obstacle was the incomplete annotation of the bovine genome, hindering the accurate characterization of single cells and their activation states. To mitigate this, we leveraged various databases and publications to identify respective genes/proteins, though many genes remain unannotated. Thus, to ensure confidence in cell clustering, we used a higher number of principal components in our PCA analysis, resulting in 10 distinct clusters identified with 28 gene markers [9,10,11,12]. Differential gene expression analysis revealed that the bMSCs skew toward a type-2 immune response with T cells expressing Th_2_-related genes, including *IL13*, *CCL1*, and *GATA3*, while macrophages exhibited high expression of matrix metalloproteinases (*MMP1*, *MMP3*, *MMP12*), which are upregulated by Th_2_ cytokines [11,51,52].

When influenza A virus enters cells, it triggers a complex cascade of immune responses, including the production and release of cytokines and chemokines. This process plays a crucial role in alerting and activating the immune system to combat the viral infection [53,54,55,56]. We observed that the number of immune cells after exposure increased substantially with upregulation of their activation, whereas the number of luminal epithelial cells (subset 2 and 3) expressing milk genes decreased. There was a considerable difference between cytokine gene expression among these subsets after exposure: with subset 3 expressing a high amount of *CSF2, IL1B,* and *TNF* genes, which corresponds to the first wave of cytokines after infection; and subset 2 cells expressing genes related to the second wave of cytokines, namely *IFNG, TGFB,* and *IL12B* [55,56].

Macrophages and DCs are among the first to respond to infection signals. They recognize common pathogen features through pattern recognition receptors (PRRs). Upon activation, they engulf pathogens and present fragments on their surface and release additional cytokines and chemokines to recruit more immune cells [53]. We observed a similar pattern with the number of both macrophages and DCs increasing during infection and upregulation of genes corresponding to their activation. These cells likely phagocytize epithelial cells, as observed by a decrease in luminal epithelial cell populations occurring within 24 h, a time frame too short for viral cytolysis. The cytokine release and antigen processing and presentation can further activate other immune cells such as B and T cells [53]. We did not observe B cells in our bMSCs; however, we observed two sets of T cells, both of which correspond to Th_2_ cells in different functional states: activated and inhibited. The number of inhibited T cells (subset 1) decreased after exposure with upregulation of Th17-related genes *RORC, IL17A,* and *IL17F*; however, downregulation of *STAT3* (Appendix A), which is crucial for the development and function of Th_17_ T cells, implies a dysregulation of these T cells [57,57,58,59]. The number of activated T cells (subset 2) increased substantially after infection with upregulation of the *IFNG, STAT3, STAT4, IL9, FOXP3*, and *TGFBR1, TGFBR2* genes. TGF-β1 induces Foxp3 expression in naive CD4+ T cells, leading to the generation of induced regulatory T cells (iTregs) [60,61,62]. Upregulation of the Foxp3 and TGF-β1 receptor genes implies that after exposure, these T cells were induced to iTregs via the actions of cytokines secreted from infected epithelial and innate immune cells. However, further experiments to explore the re-programming of T cells is required.

Overall, our in vitro analysis highlights the diverse host immune responses triggered by bovine H5N1 exposure in different cell populations, particularly the upregulation of both pro-inflammatory and regulatory pathways in response to infection. While different epithelial cell subsets in bMSCs exhibited varying transcriptional responses to H5N1 exposure, we did not detect viral RNA reads in the scRNA-seq data, and thus cannot confirm productive infection at the single-cell level. Although qRT-PCR targeting the M gene detected viral RNA in bulk bMSC samples, this may reflect exposure to viral components rather than intracellular replication. The specific cell types permissive to viral entry or replication remain unclear and warrant further investigation. Future studies involving cell sorting of individual bMSC subsets followed by qRT-PCR or viral antigen detection would be instrumental in identifying the precise cellular targets of H5N1. These findings underscore the importance of continued research into the mechanisms of H5N1-induced inflammation and mastitis, and support the need for the development of targeted prevention and therapeutic strategies against this emerging threat.

## 5. Limitations

This study has several limitations. First, only one scRNA-seq library per condition was generated due to resource constraints and the need for high-containment BSL3 facilities, which limits our ability to evaluate biological variability. Second, bovine milk somatic cells (bMSCs) are a heterogeneous and predominantly non-proliferative population with limited viability ex vivo, which may reduce their suitability for modeling productive HPAIV infection. Third, neutrophils—typically a major cell type in bovine milk—were not detected in our dataset, likely due to their short lifespan (less than 24 h) and potential loss during sample processing. Finally, although qRT-PCR detected viral RNA in H5N1-exposed bMSC samples, the absence of viral reads in the scRNA-seq dataset suggests that these cells may not support active viral replication. As such, the observed transcriptional responses are most likely driven by exposure to viral components rather than definitive evidence of productive infection.

## Figures and Tables

**Figure 1 viruses-17-00811-f001:**
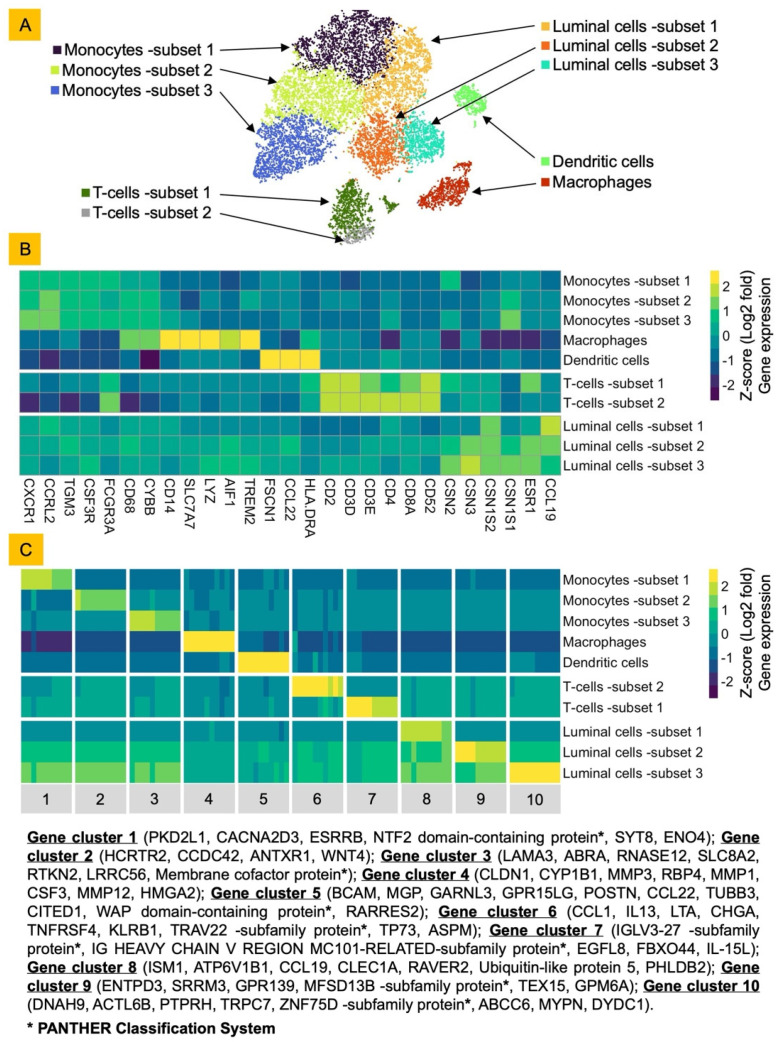
Single-cell RNA-seq analysis of bovine milk somatic cells (bMSCs). (**A**) t-SNE embeddings of merged scRNA-seq profiles from in vitro-unexposed and bovine H5N1-exposed bMSCs collected from three healthy cattle. Distinct cell populations are represented by different colors and assigned labels based on expression of selected gene markers. (**B**) Heatmap showing z-scored average expression of curated cell marker genes that had a fold change >1 and *p* < 0.05 for at least one cluster. Genes are ordered by the cluster in which they have the highest expression. (**C**) Heatmap displaying z-scored average expression of top 10 genes in each cluster that had a fold change >1 expressed and *p* < 0.05 for at least one cluster. Genes are organized by cluster in which they have highest expression.

**Figure 2 viruses-17-00811-f002:**
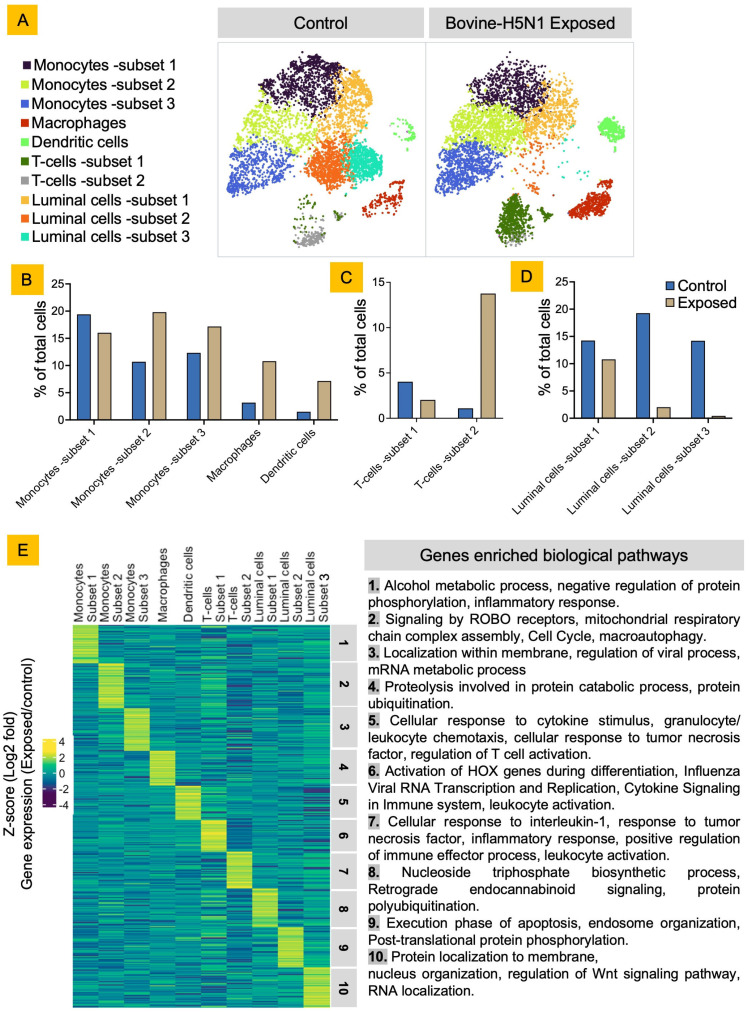
Comparison of cell populations and gene expression between control and bovine H5N1-exposed bMSCs. (**A**) t-SNE embeddings of bMSCs either exposed to bovine H5N1 or unexposed control cells. Distinct cell populations are represented by different colors and assigned labels based on expression of selected gene markers, same as in Figure 1. (**B**–**D**) Bar graphs showing percentage (x-axis) of innate immune (**B**), T (**C**), and epithelial (**D**) cells in total cells, corresponding to different cell types (y-axis) in control (blue) and bovine H5N1-exposed (brown) bMSCs. (**E**) Heatmap displaying z-scored average expression of top 50 upregulated genes after bovine-H5N1 infection in each cluster that had a fold change >1 expressed and *p* < 0.05 for at least one cluster. Genes are organized by cluster in which they have highest expression and analyzed for Gene Ontology (GO)-enriched processes.

**Figure 3 viruses-17-00811-f003:**
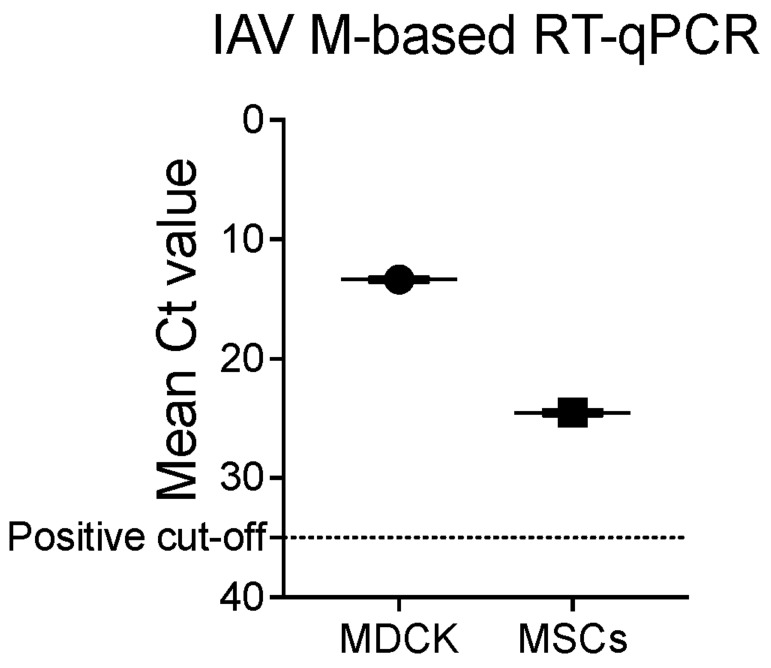
Viral genome detection by RT-qPCR. The graph shows the results of IAV M gene-based RT-qPCR in MDCK cells (positive control) and bMSCs infected with bovine H5N1. The x-axis represents the mean Ct values, and y-axis represents the cell type.

## Data Availability

The original data presented in the study are openly available in NCBI’s Gene Expression Omnibus (GEO) at accession number GSE294077.

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
