# Peer review of "Single-Cell Analysis of Host Responses in Bovine Milk Somatic Cells (bMSCs) Following HPAIV Bovine H5N1 Influenza Exposure"

_viruses, 2025, doi:10.3390/v17060811_

Round 1
Reviewer 1 Report
Comments and Suggestions for Authors
Summary
This manuscript explores immune responses in bovine milk somatic cells (bMSC) following in vitro exposure to a 2024 U.S. dairy farm isolate of H5N1 HPAIV, using scRNA-seq. The study is timely and addresses a pressing concern in veterinary and public health. The work aims to characterize immune and epithelial cell responses to viral exposure, with a focus on cytokine profiles and cell type shifts. While the study is potentially informative, several key issues undermine the strength of the conclusions.
Major Weaknesses
The manuscript repeatedly refers to "infected" cells, implying productive viral infection. However, the authors do not report any detection of infectious virus or viral RNA reads in the scRNA-seq data. Without confirmation of viral presence, replication, or entry into the cells, the observed responses cannot be definitively linked to infection. No plaque assay, TCID50, or sequencing-based detection of viral transcripts is presented to validate infection. Given the central role this claim plays in the manuscript, this omission is critical. The term “infection” should be reserved for cases where viral replication is confirmed, otherwise, the results may reflect a response to viral antigens or inactivated virus rather than active infection.
The observed immune signatures (IL13+ T cells, macrophage shifts) are interesting but must be interpreted in the context of viral exposure rather than infection. I believe, the conclusions drawn about host-pathogen interaction, viral immune modulation, and implications for therapeutic strategies are therefore overstated in their current form. To assist this i suggest the authors include details of the degree of variability between mock-infected replicates. Were there any differentially expressed genes (DEGs) between mock samples that could suggest batch effects or influence interpretation for example?
Furthermore, the authors should clarify the limitations of bovine milk somatic cells as a model system. These cells are a mix of short-lived immune and epithelial cells, typically not optimized for culture or viral replication studies. Without evidence that these cells can proliferate, survive in cell cultire and support H5N1 infection —or that they are infected in vivo—the conclusions about viral pathogenesis are speculative. The study would benefit from either a more permissive in vitro system (e.g., bovine mammary epithelial cell lines) or direct demonstration of viral RNA/proteins within these cells during exposure.
Author Response
This manuscript explores immune responses in bovine milk somatic cells (bMSC) following in vitro exposure to a 2024 U.S. dairy farm isolate of H5N1 HPAIV, using scRNA-seq. The study is timely and addresses a pressing concern in veterinary and public health. The work aims to characterize immune and epithelial cell responses to viral exposure, with a focus on cytokine profiles and cell type shifts. While the study is potentially informative, several key issues undermine the strength of the conclusions.
Response: We sincerely thank the reviewer for their constructive feedback and insightful suggestions. We have carefully revised the manuscript to address all concerns. Below, we provide a point-by-point response to each comment. All changes are highlighted in the revised manuscript.
Major Weaknesses
- The manuscript repeatedly refers to "infected" cells, implying productive viral infection. However, the authors do not report any detection of infectious virus or viral RNA reads in the scRNA-seq data. Without confirmation of viral presence, replication, or entry into the cells, the observed responses cannot be definitively linked to infection. No plaque assay, TCID50, or sequencing-based detection of viral transcripts is presented to validate infection. Given the central role this claim plays in the manuscript, this omission is critical. The term “infection” should be reserved for cases where viral replication is confirmed, otherwise, the results may reflect a response to viral antigens or inactivated virus rather than active infection.
Response We agree with the reviewer’s comment. We have replaced "infection" with "exposure" throughout the revised manuscript, except in cases where qRT-PCR data are discussed. We have also included clarifying statements in the Abstract, Results, and Discussion to explain that no viral RNA was detected by scRNA-seq, and the observed responses likely reflect exposure rather than productive infection.
- The observed immune signatures (IL13+ T cells, macrophage shifts) are interesting but must be interpreted in the context of viral exposure rather than infection. I believe, the conclusions drawn about host-pathogen interaction, viral immune modulation, and implications for therapeutic strategies are therefore overstated in their current form. To assist this i suggest the authors include details of the degree of variability between mock-infected replicates. Were there any differentially expressed genes (DEGs) between mock samples that could suggest batch effects or influence interpretation for example?
Response: We agree with the reviewer’s comment. We have revised the Abstract and Discussion to adjust the conclusions accordingly and avoid over-interpretation. Specifically, we now clarify that the immune responses observed may result from exposure to viral components rather than confirmed productive virus infection. We appreciate the reviewer’s suggestion regarding the importance of evaluating variability, particularly given the potential for undetected exposure or infection during sample collection. However, due to resource constraints, samples were pooled prior to sequencing, which precluded the ability to assess batch effects or inter-sample variability. We have acknowledged this limitation in the newly added Limitations section of the manuscript.
- Furthermore, the authors should clarify the limitations of bovine milk somatic cells as a model system. These cells are a mix of short-lived immune and epithelial cells, typically not optimized for culture or viral replication studies. Without evidence that these cells can proliferate, survive in cell cultire and support H5N1 infection —or that they are infected in vivo—the conclusions about viral pathogenesis are speculative. The study would benefit from either a more permissive in vitro system (e.g., bovine mammary epithelial cell lines) or direct demonstration of viral RNA/proteins within these cells during exposure.
Response: We agree with the reviewer’s comment. We have added a Limitations section to the revised manuscript acknowledging that bMSCs are short-lived and may not be optimal for modeling productive viral replication. To address this limitation, we are currently conducting follow-up studies using bovine and other species-derived epithelial cell lines to gain deeper insights into H5N1 pathogenesis.
Reviewer 2 Report
Comments and Suggestions for Authors
The manuscript tackles an important, yet understudied, aspect of bovine immunology by profiling milk‑derived somatic cells (bMSCs) following experimental H5N1 exposure. The single‑cell approach is timely and has clear relevance. Several issues need to be addressed, as detailed below.
- Cell‑types. The authors report monocytes/macrophages, T cells, dendritic cells and non‑immune epithelial cells, but no neutrophils. This is unexpected: neutrophils typically constitute a major cell population in bovine milk. Can you please explain the cause of this —e.g., by detailing sample handling, filtration or selection that was applied, or marker choices.
- Choice of sampling time point. The rationale for the selected time point is unclear. Viral‑load kinetics (e.g., plaque assay or qRT‑PCR for the M segment) across multiple early time points would justify the selected window and help interpret the transcriptional state captured by scRNA‑seq.
- Biological replication. Only one scRNA‑seq library per condition was generated. While single‑cell datasets contain many cells, they still represent a single biological replicate or a single pool of replicates. If additional bMSC samples are unavailable, bulk RNA‑seq on bMSC could provide support for the reported results. Please add such support if there are published bulk transcriptome datasets of bMSC samples. If such data is not available, please comment on this that there was no published data for additional comparisons.
- Cell‑quality bias between conditions. Infected cells often exhibit higher mitochondrial reads and lower UMI counts, confounding differential‑expression (DE) analyses. The manuscript can gain from reporting key QC metrics (e.g., median genes /cell, % mitochondrial reads) for each condition. In addition, it would be valuable to report the DE results after regressing out low‑quality cells in the revised manuscript.
- Evidence of productive replication. No viral reads were detected in the poly‑A–selected libraries, yet qRT‑PCR targeting the M gene was positive. Because qRT‑PCR does not discriminate between intracellular genomic RNA and cell free virions, the data do not confirm productive replication—or even primary viral transcription—within bMSCs. The authors should clarify whether the virus truly enters and transcribes in these cells, and the observed host response reflects genuine infection, or otherwise, indicate that this is not clear. If additional data cannot be added, then at least this point should be clarified in the revised manuscript.
Author Response
The manuscript tackles an important, yet understudied, aspect of bovine immunology by profiling milk‑derived somatic cells (bMSCs) following experimental H5N1 exposure. The single‑cell approach is timely and has clear relevance. Several issues need to be addressed, as detailed below.
Response: We sincerely thank the reviewer for their constructive feedback and insightful suggestions. We have carefully revised the manuscript to address all concerns. Below, we provide a point-by-point response to each comment. All changes are highlighted in the revised manuscript.
1. Cell‑types. The authors report monocytes/macrophages, T cells, dendritic cells and non‑immune epithelial cells, but no neutrophils. This is unexpected: neutrophils typically constitute a major cell population in bovine milk. Can you please explain the cause of this —e.g., by detailing sample handling, filtration or selection that was applied, or marker choices.
Response: We agree with the reviewer’s comment. We believe the absence of neutrophils in our dataset is likely due to their short lifespan and their removal during the dead cell filtering steps. Additionally, neutrophil numbers may have further declined during the 24-hour incubation period. This limitation has been acknowledged in the newly added Limitations section of the revised manuscript.
2. Choice of sampling time point. The rationale for the selected time point is unclear. Viral‑load kinetics (e.g., plaque assay or qRT‑PCR for the M segment) across multiple early time points would justify the selected window and help interpret the transcriptional state captured by scRNA‑seq.
Response: The 24-hour post-exposure time point was selected based on viral kinetics observed in MDCK cells, which are permissive to influenza A virus replication. At this stage, viral replication typically peaks without significant cell death, making it optimal for detecting host transcriptional responses while maintaining cell viability—a key requirement for scRNA-seq protocols. This rationale has been added to the revised Methods and discussed in the manuscript (Line 184-186).
3. Biological replication. Only one scRNA‑seq library per condition was generated. While single‑cell datasets contain many cells, they still represent a single biological replicate or a single pool of replicates. If additional bMSC samples are unavailable, bulk RNA‑seq on bMSC could provide support for the reported results. Please add such support if there are published bulk transcriptome datasets of bMSC samples. If such data is not available, please comment on this that there was no published data for additional comparisons.
Response: We agree with the reviewer that additional biological replicates would strengthen the confidence in our findings. However, due to resource constraints and limited access to high-containment BSL-3 facilities, we were only able to process a limited number of samples for this study. This limitation has been acknowledged in the newly added Limitations section in the revised manuscript. We are currently conducting follow-up studies using bovine and other species-derived epithelial cell lines to further investigate H5N1 pathogenesis in cattle. As part of our ongoing efforts, we are also generating additional data that will be made publicly available to support further research and comparative analyses.
- Cell‑quality bias between conditions. Infected cells often exhibit higher mitochondrial reads and lower UMI counts, confounding differential‑expression (DE) analyses. The manuscript can gain from reporting key QC metrics (e.g., median genes /cell, % mitochondrial reads) for each condition. In addition, it would be valuable to report the DE results after regressing out low‑quality cells in the revised manuscript.
Response: We agree with the reviewer’s comment. We have now clarified in the reised manuscript (Line 111) that cells with fewer than 5,000 reads—representing low-quality cells—were removed prior to downstream analysis. All reported results are based on the filtered dataset. Additionally, we note in the revised manuscript (Line 134) that the mean reads per cell were approximately 16,500. Due to the incomplete annotation of the bovine genome, we were unable to reliably quantify mitochondrial read content, as current tools are optimized primarily for human and mouse genomes. This limitation has been acknowledged in the Discussion (Line 275).
4. Evidence of productive replication. No viral reads were detected in the poly‑A–selected libraries, yet qRT‑PCR targeting the M gene was positive. Because qRT‑PCR does not discriminate between intracellular genomic RNA and cell free virions, the data do not confirm productive replication—or even primary viral transcription—within bMSCs. The authors should clarify whether the virus truly enters and transcribes in these cells, and the observed host response reflects genuine infection, or otherwise, indicate that this is not clear. If additional data cannot be added, then at least this point should be clarified in the revised manuscript.
Response: We agree with the reviewer’s comment and added clarification on lines 192-195 in our revised manuscript.
Reviewer 3 Report
Comments and Suggestions for Authors
The manuscript by Singh et al. describes the scRNA-seq analyses of bMSCs infected by HPAIV bovine-H5N1 influenza virus. Their findings showed that bovine-H5N1 infection can trigger complex immune responses. Overall, this study is extremely important and deserves to be published.
Major concern:
- A major question about this study is which group of cells in bMSCs could be infected by H5N1. If the authors could perform cell sorting to separate each cluster of cells in bMSCs and then detect H5N1 in each cluster of cells by qRT-PCR. The results maybe it would be able to provide some clues about the exact cells infected by bovine-H5N1. At least the supplementary figure 1 should be moved to the main text of this manuscript.
Minor concern:
- Line 59, “SPF free chicken egg” should be “SPF chicken egg”.
Author Response
The manuscript by Singh et al. describes the scRNA-seq analyses of bMSCs infected by HPAIV bovine-H5N1 influenza virus. Their findings showed that bovine-H5N1 infection can trigger complex immune responses. Overall, this study is extremely important and deserves to be published.
Response: We sincerely thank the reviewer for their constructive feedback and insightful suggestions. We have carefully revised the manuscript to address all concerns. Below, we provide a point-by-point response to each comment. All changes are highlighted in the revised manuscript.
Major concern:
- A major question about this study is which group of cells in bMSCs could be infected by H5N1. If the authors could perform cell sorting to separate each cluster of cells in bMSCs and then detect H5N1 in each cluster of cells by qRT-PCR. The results maybe it would be able to provide some clues about the exact cells infected by bovine-H5N1. At least the supplementary figure 1 should be moved to the main text of this manuscript.
Response: We acknowledge the value of this suggestion. While we were unable to sort and analyze specific cell types due to the limited availability of cattle immune cell antibodies and the limitations working in a high containment BSL-3 facility, we have included this as a future direction in the Discussion.
Minor concern:
- Line 59, “SPF free chicken egg” should be “SPF chicken egg”.
Response: We have corrected "SPF free chicken Egg" to "SPF chicken egg" in the Methods section.
Round 2
Reviewer 1 Report
Comments and Suggestions for Authors
improvements are satisfactory